# Automatic Equipment to Increase Sustainability in Agricultural Fertilization

**Mario Martínez García** [1], **Silvia Ramos Cabral** [1], **Ricardo Pérez Zúñiga** [2,*]
and **Luis Carlos G. Martínez Rodríguez** [1]

1. Department of Computer Science and Engineering, Campus CUValles, Universidad de Guadalajara, Ameca Highway Km. 45.5, P. C., Ameca 46600, Jalisco, Mexico
2. Virtual University System Department, Universidad de Guadalajara, Av Enrique Díaz de León Sur 782, Moderna, Guadalajara 44200, Jalisco, Mexico
* Correspondence: perezuniga@udgvirtual.udg.mx; Tel.: +52-3328100572

**Abstract:** The purpose of the study was to design a device for the dissolution of fertilizers for agricultural use in an automatic and environmentally sustainable way to facilitate the work of farmers. To achieve this goal, an outdated blade design was used, which generates turbulent and laminar flows thanks to the angle of inclination of its blades. In tests, the combination of these two flows gave a better result compared to laminar and turbulent flows separately. The best results were achieved by varying the spin and speed, the time between spins, and the rest time. The time it would take to dissolve the mixture was drastically reduced if it were conducted in the traditional way (manually) or compared with commercial mixers. In conclusion, the technique used for the dissolution of agricultural minerals is more effective and reduces time, energy, and effort. This was able to reduce the time necessary to dissolve the fertilizer by 93 percent compared to doing it manually and by 66 percent compared to using commercial mixers, in a solution of 100 L of water per 100 kg of ammonium sulfate.

**Keywords:** automatic equipment; fertilizer dissolution; agriculture; machinery; automation; energy saving; beater

## 1. Introduction

This article presents an automatic equipment to dissolve agricultural fertilizers to apply them as foliar or directly to the soil and improve the health of the plants; according to Allen, the fortification of plants with nutrients can have positive effects on the health of those who consume them [1,2]. The design is based on a pair of found blades propelled by a motor with a card that controls rotation in both directions to avoid spilling the nutrients out of the tank where they are dissolved, in addition to a suction pump for timely access. The main objective of the present invention is to provide a system for dissolving agricultural nutrients and/or fertilizers for their subsequent application to crops, which allows reducing the time needed to dissolve the components of the mixtures to be used. This fertilization technique is used in some agricultural areas to reduce the loss of nutrients, because by liquid fertilizing the plants immediately after the application of the nutrients begin with their use and begin to make the distribution of nutrients more efficient and sustainable. regarding the methods using application of solid fertilizers [3–6].

In addition, liquid fertilization reduces toxicity and the risk of contaminating the environment [7,8] because the nutrients are absorbed directly in the application area and immediately used by the target crops. The proposed mechanism consists of a frame to protect the motor, a system to cushion the generated forces, a level adjustment system, a base to support the structure in the container for dissolving the fertilizers, a direct current motor with a nominal power of 250 watts, a speed of 3000 revolutions per minute, and a nominal torque of 0.80 newton/meter. The motor is powered by a current of 12 volts

and 13 amps and has a mechanical gear reduction of 9.78 turns to one, which is necessary to be able to support the load it will be subjected to during operation. The mechanism also includes a control card, a suction pump, and two flow vanes for dissolving fertilizers and/or agricultural nutrients, positioned strategically for proper dissolution; one of the flow vanes is positioned to generate a turbulent flow when advancing through the fluid, and the second flow vane is positioned to generate a laminar flow [9,10]. The purpose of these two flows is to meet and then combine to generate greater turbulence, which would collide with the solute particles and cause friction to dissolve the substance more quickly [11,12].

The blades are positioned at certain angles in turn. The first blade has an angle of ninety degrees relative to the flow of the substance, which reduces lift force. The second blade has an angle of zero degrees relative to the flow of the substance, which also reduces lift force. Because no extra force is generated, no additional force is exerted on the structure [13,14].

Agriculture is a vital part of our way of life, helping to reduce poverty and promoting the sustainability of our society [15–17]. It is important to support farmers with technological advancements that make their work easier and have less impact on the environment [18,19]. However, in Latin American countries such as Mexico, there is little support for new technologies and methods that can make food production more efficient and reduce costs in plant fertilization [20,21]. Despite this, farmers must have access to methods and techniques that allow them to manage the application of fertilizers in their crops to reduce nutrient losses due to evaporation and other chemical processes and to improve production [2,7,22]. Currently, the agriculture sector is being affected by the increasing costs of fertilizers, which have been impacted by the increase in the costs of fertilizers that have been affected by the COVID-19 pandemic and restrictions [23–27], as well as the invasion of Russia into Ukraine and the increase in the price of gas, the latter being essential in the production of nitrogen in the form of ammonia and subsequently dehydrated into urea; the main producers of this fertilizer in the world are Russia, China, and India, and they have imposed restrictions on their exports to ensure internal supply [28].

The world needs to produce food; countries need to ensure food production for their people [29,30]; however, they do little or nothing to ensure the supplies that farmers need to maintain food production, relying on imports and dependence on third parties, which puts the viability of food production at risk, and adding to this the inefficiency in the application of nutrients such as the traditional solid and volatilized method to crops generates losses of fertilizers due to evaporation, plant competition, microorganisms, and water wash-off in crops [31].

Nutrient loss is something that farmers need to reduce; especially with the high costs of fertilizers, it is necessary to find value-added methods that allow for better use of plant nutrients, such as composts, leachates, and liquid fertilization as promoted by the invention of this article, reducing nutrient loss and being environmentally friendly [32], reducing environmental pollution, reservoirs, and water reserves [33,34]. Current consumers demand food that is produced sustainably, safely, and transparently. To meet these demands, farms and companies need to improve their production [35]. To meet this reality, other methods of fertilizer application, meat, and food production [36] are required, and there are alternatives such as foliar nutrition and liquid fertilization.

The agricultural sector, among others, is not yet fully automated, mainly in Latin American and Caribbean countries due to their characteristics. This is why manual mixing or dissolutions are still carried out, primarily when the mixing action needs to be carried out on site where there is no way to transport an industrial mixer or even there is no source of electricity. This requires more time and becomes more laborious and costly due to the man-hours involved. An example of these tasks is the mixing or dissolution of two or more solid–liquid, liquid–liquid, or solid–solid fertilizers, a task that is usually carried out using a manual or electric mixer, or the farmer even uses a "log" to obtain the mixture. Portable electric mixers or blenders, such as for mixing cement or paint, use alternating current to supply the electrical demand generated by the mixer or blender machine; they usually

rotate in one direction; their weight is approximately 7 kg; and their length varies between 24 and 60 cm, making it difficult to use in the fields as well as their electrical supply because this type of work is carried out at distances far from an electrical outlet.

This article is organized as follows. In Section 2, a brief explanation and diagrams of materials and methods, Section 2.1 electronic stage, and Section 2.2 the logic sequence of the software to control the turns are shown; in Section 3 the optimal results for dissolving the mixtures in the shortest time are shown; in Section 4 the discussion is presented; and finally, conclusions are presented in Section 5.

## 2. Materials and Methods

### 2.1. Tool Development

The mechanical stage of the proposed device is composed of a rectangular frame which serves to protect and fix the motor; said frame is composed of four metal sills positioned at different levels, four fixing rods which in turn serve to dampen the forces generated and that could lead to misaligning the rotor head due to some misuse of the device, generating protection for the user.

On the other hand, Figure 1 shows a side view with separated parts where the design of the proposed structure can be appreciated. Figure 2 shows a real view of the proposed mechanical device.

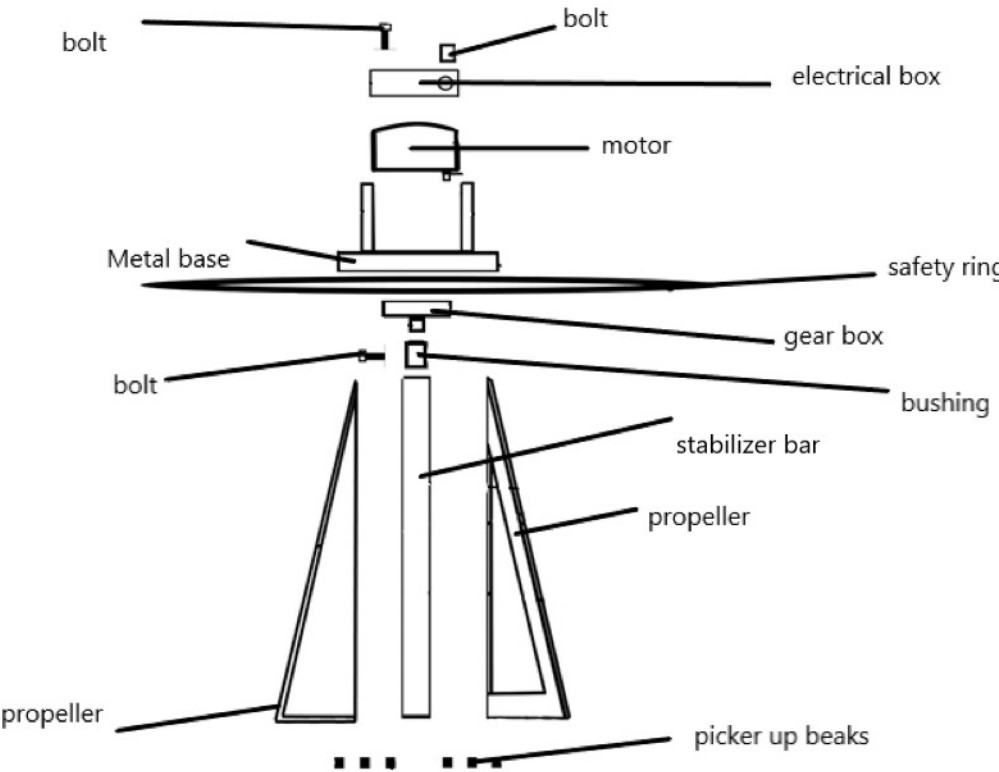

**Figure 1.** Side view with separate parts.

The fixing rods, also called guide rods, are mounted on a rectangular base and have a leveling adjustment system, in case it is necessary to level the automatic equipment due to the lack of space that may be generated at the time of its installation.

The rectangular base on which the guide rods and the frame are mounted is supported by four arms, which in turn are supported on the upper face of a thirty-centimeter radius metallic ring which embraces a container in which the mixture of agricultural salts and solvent will be found; said container has a diameter of 60 cm and an approximate height of eighty centimeters, generating a volume of approximately two hundred liters.

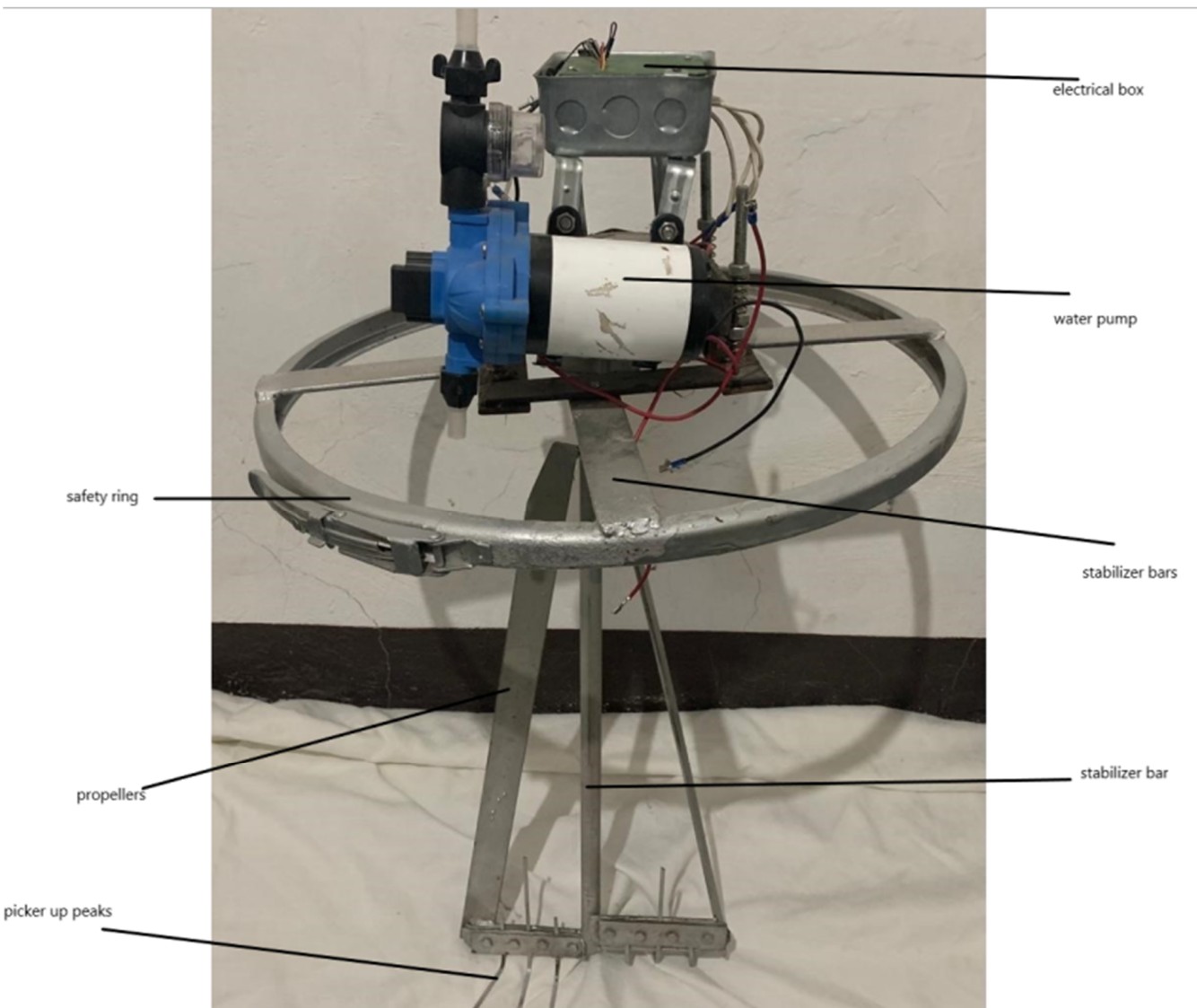

**Figure 2.** Real view of the mechanical device.

At the bottom of the head are the lifting peaks whose function is to lift the remains of the mixture that fell to the bottom of the container due to their size, thus avoiding damage to the structure when the automatic equipment for dissolving agricultural salts is lowered. The lifting peaks have an adjustment so that the user can position them in the most optimal place.

It has a head consisting of a perforated stainless steel tube to prevent corrosion of the substances that will be used. On the opposite sides of the circumference of the tube are two flow paddles made of stainless steel, positioned strategically for correct dissolution; one of them is positioned in such a way that when it moves through the fluid, it generates a turbulent flow; the second flow paddle is positioned in such a way that it generates a laminar flow. The purpose of these two flows is to meet and then combine to generate greater turbulence, which would collide with the particles of the solute, causing the substance to dissolve more quickly. The paddles are also positioned at certain angles: the first has a 90-degree angle on the flow of the substance, resulting in a total loss of lift force, and the second has a zero-degree angle on the flow of the substance, which also generates a loss of lift due to the lack of this force. No extra force will be applied to the structure.

In Figure 3, the top view of the proposed device, the control head can be seen, which belongs to the main base system is a metal box that stores a plastic cubic structure inside as shown in Figure 4 (this is to avoid contact with the high voltage parts of the chassis and prevent the circuits from being damaged) in which the circuits responsible for the electrical control of the motor are located.

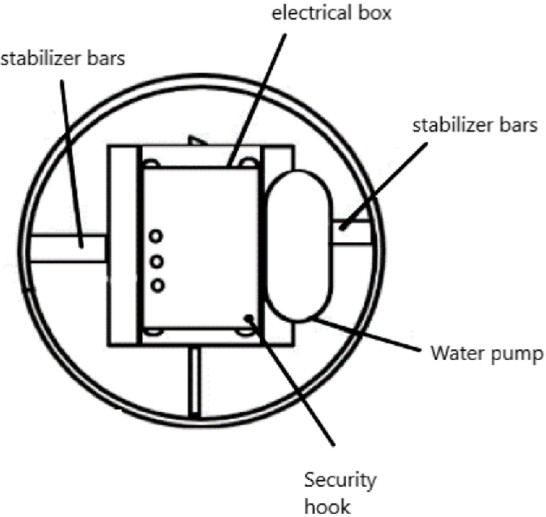

**Figure 3.** Top view.

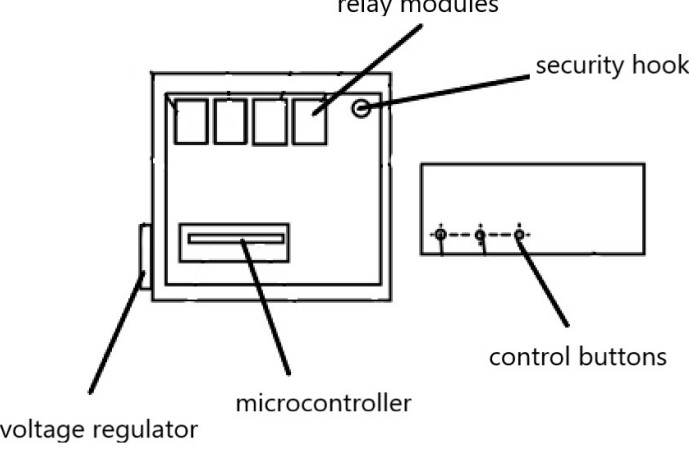

**Figure 4.** Internal view where the control system is located.

The engine that is part of the main base assembly has a gearbox assembled on the output shaft as shown in Figure 5, which is responsible for increasing the final mechanical force that will be directed to the propellers, in order to improve the efficiency of the automatic equipment and increase sustainability in agricultural fertilization.

The main base assembly-propellers are connected using a hub that extends from the gearbox shaft in conjunction with a presser to secure the structure.

The present invention has a direct current motor which has an external metal structure with a rectangular base, which is positioned on the main frame; this base is assembled to the main structure helping to support the weight of the head. The motor has a nominal power of 250 watts, a speed of 3000 revolutions per minute, and a nominal torque of 0.80 newton/meter; the motor is powered by a current of 12 volts and 13 amps; it has a mechanical gear reduction of 9.78 turns per one, which is necessary to be able to support large loads to which it will be subjected during work.

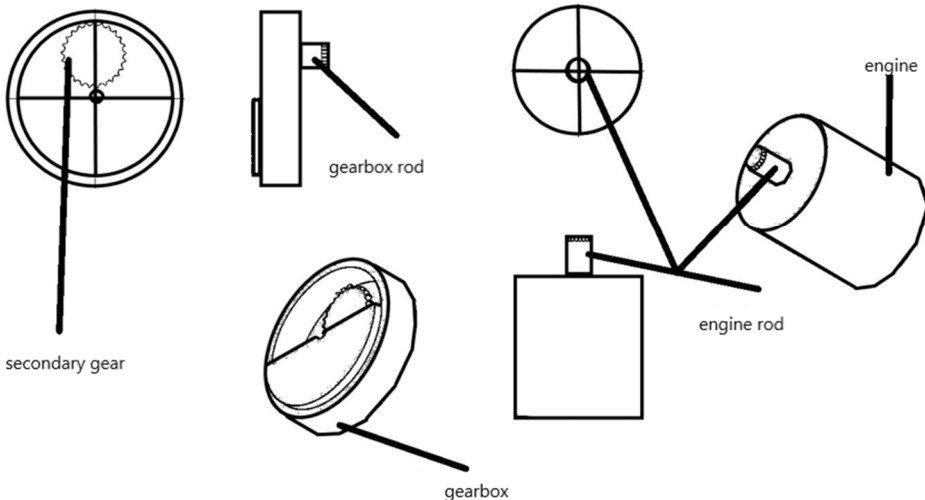

**Figure 5.** Views of the motor and transmission.

On the output bar of the geared motor is an adapter which allows the motor's twisted output to be connected to the head's input without damaging the structure of the output bar; this adapter is screwed onto the output bar and secured with a pressure nut, thus avoiding any imbalances of the head during maintenance.

In Figure 6 it is noted that the connection diagram between the microcontroller and some of the rotation activation components such as relays and connection pins for the input signals, which are processed and then sent to the microcontroller for analysis, can be seen.

### 2.2. Electronic Stage

On the main base are the electronic components that control the entire mechanical system, also called the control head, which consists of a microcontroller powered by a voltage regulator which is connected to a 12-volt output. There is a system of 4 relays in series to control the movements of the motor, in addition to a button center, which has three functions: turn on, start cycle, and turn off in that respective order.

The microcontroller works in conjunction with a 16 MHz quartz oscillator in parallel with two 22 picofarad capacitors. The microcontroller receives information from the buttons and activates the relays to start the cycle. It is recommended to place a diode at the input of each relay to prevent voltage spikes towards the microcontroller.

In this implementation, the ATMEGA328p microcontroller was used. This last one meets the specifications mentioned above.

As shown in Figure 6, four relay modules are used which work together with a microcontroller with five analog input/outputs and 14 digital inputs/outputs, of which four analog outputs and 10 digital inputs/outputs are used with the ability to activate internal resistors. Four of the analog outputs will be connected to the relays; three analog outputs will be connected to three pushbuttons; three analog outputs will be the return of the pushbuttons; and the digital input will be responsible for measuring feedback; all of this powered by a 5-volt output voltage regulator to avoid damaging the controller in addition to the use of 12–24 voltage regulators at the motor's current input.

A diagram of the operation of the "Automatic Equipment" rotation system is shown in Figure 7. This process starts by sending 5 v pulses to the base of the transistors. This pulse sending is segmented into two stages in which in each of these stages two analog outputs send a 5-volt signal with a pulse width of 10 s and a frequency of 0.05 Hz. In the same way, the second stage emits the same pulses but with the difference that this last one is phased out with respect to the first by 10 s in such a way that the first pulse is in a high state and the second in a low state (Figure 8), and in this way the passage of current in two directions is controlled.

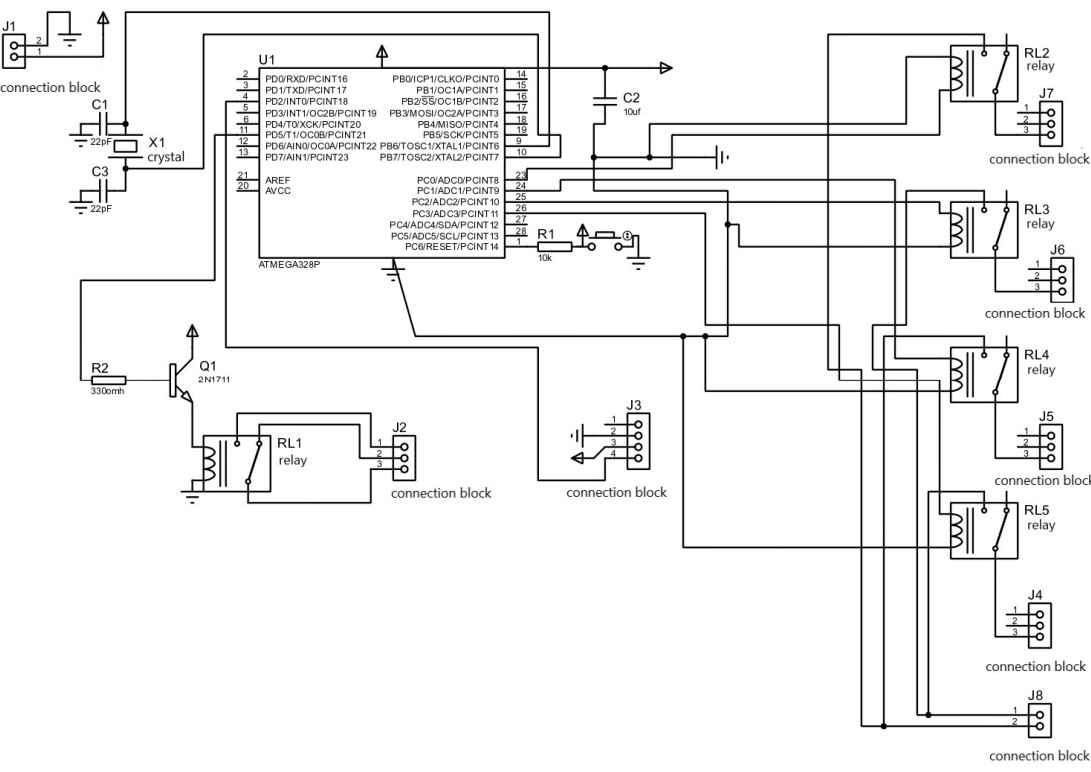

**Figure 6.** Control circuit for the mechanical system responsible for processing signals, emitting pulses, controlling relays, and switching operating mode.

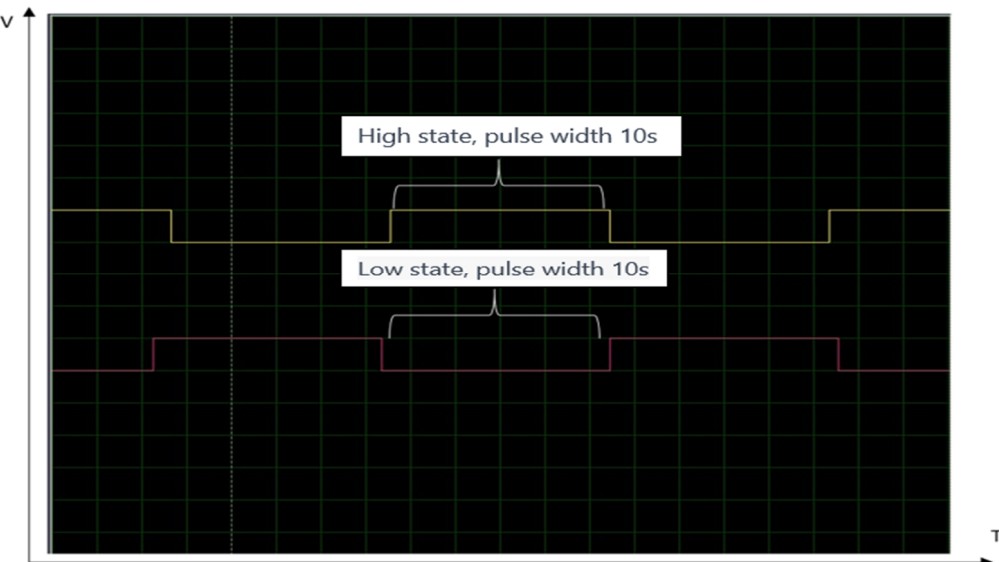

**Figure 7.** Analog outputs phased in time to control the direction of rotation of the motor.

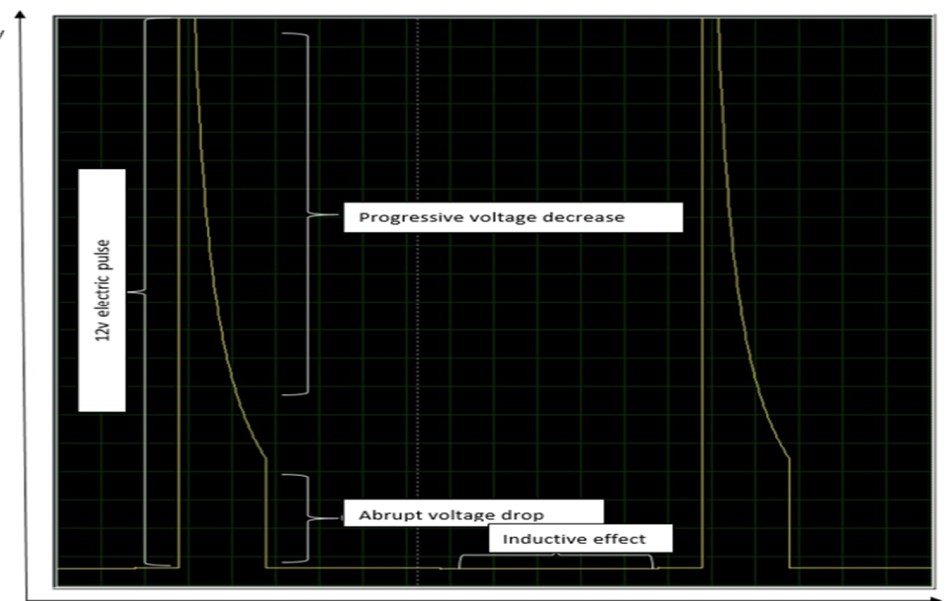

**Figure 8.** Behavior of the electric pulses sent to the motor.

This arrangement in times allows us to alternate the activation of the control stages, leaving us with an alternating rotation of the motor.

Once the electric pulse drops to 0-volt, electrical feedback is used to activate the electric brake in order to generate more turbulence in the fluid.

The electric pulse used to determine the activation of the electric brake is generated during the time interval between two pulses when deactivating the current for the change in rotation, taking advantage of the inductive effect of the motor's coils which, for a brief period of time, functions as a generator, which we take advantage of as a signal, as shown in Figure 9.

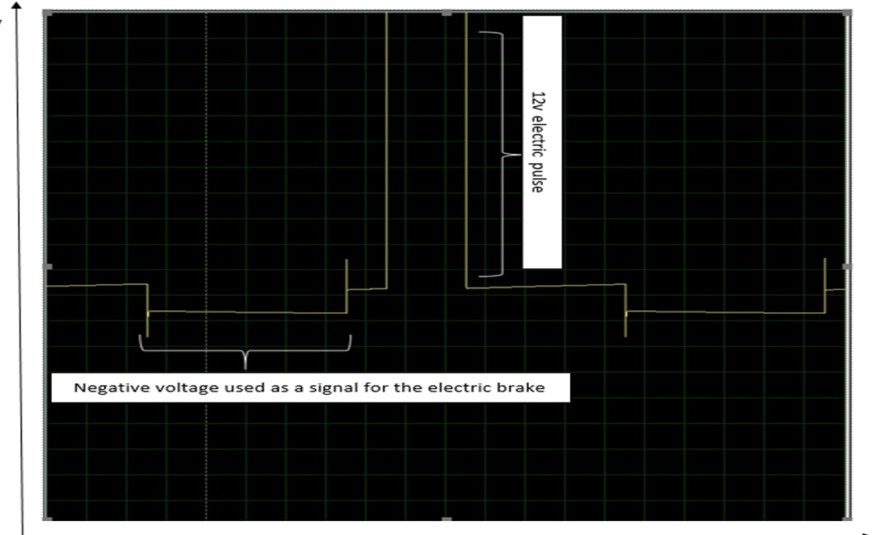

**Figure 9.** Inductive effect of the motor when power source is disconnected at a scale of 1-volt.

As seen in Figure 10, the peak voltage is relatively low, approximately −2-volt as a peak at disconnection and −1-volt for a period of 3 s before experiencing another peak of tension at the connection of approximately 2.2-volt, all due to inductive loads.

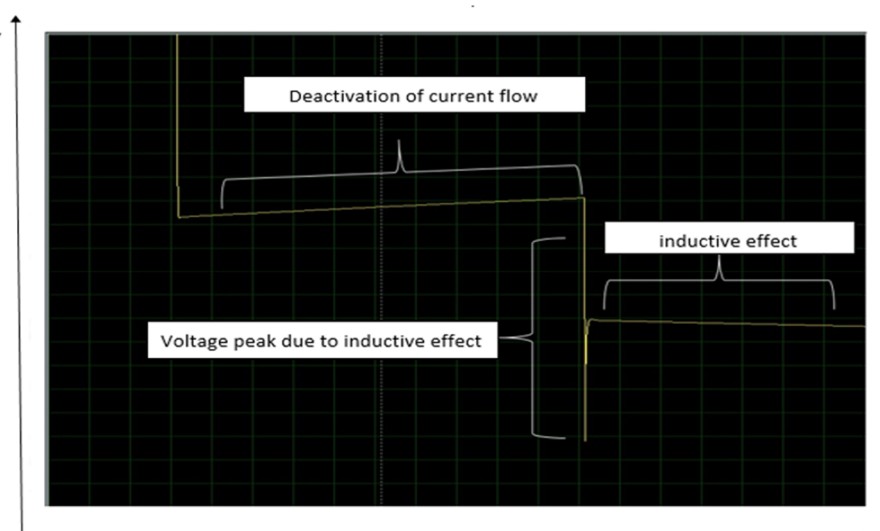

**Figure 10.** Peak voltage due to inductive effect of motor.

The electric brake consists of supplying a high voltage to the motor to make it rotate for a short period of time in the opposite direction of the original rotation once a signal emitted by a disconnection of the motor is read; this voltage is applied to the motor inputs.

As seen in Figures 11 and 12, we have two pulses working with a time shift. The first is the electric brake while the second functions as a damper to prevent the starting force of the motor in the fluid.

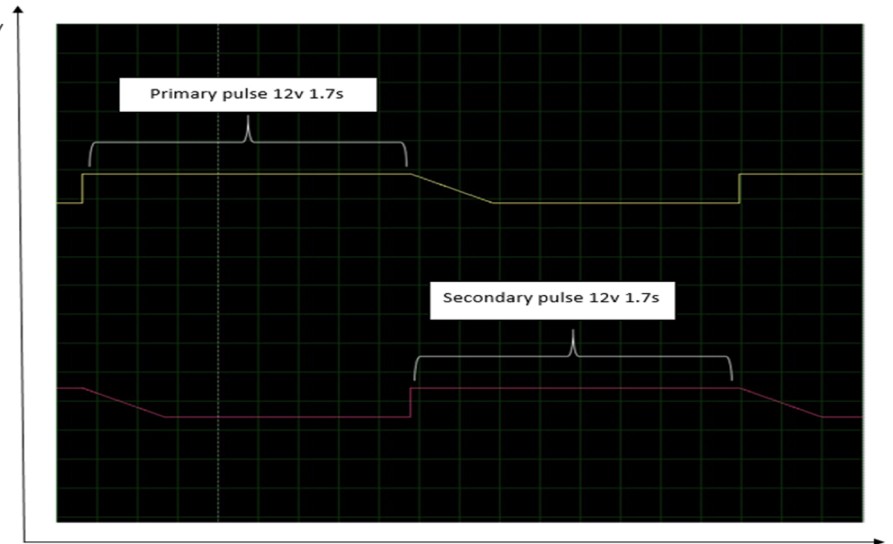

**Figure 11.** Test reading for electric brake pulse on a 12 v scale per square and 2 s.

Figure 13 shows a diagram with the steps and logical decisions taken by the controller to active the rotation and electric brake.

The controller of Automatic Equipment consists of the following components:

- Atmega328p, Microcontroller responsible for signal analysis and pulse sending.
- Four relay modules, used to alternate the flow of current.
- A 12–24v electric motor, responsible for generating the mechanical force for dissolution.
- ACS712 current sensor, used to interpret signals emitted by the motor.
- DC0-25V voltage sensor, module responsible for interpreting signals emitted by the motor.
- Four 2N1711 transistors, allowing the passage of 12-volt alternating the paths to avoid short circuits Figure 14.

- Four 2N2222 transistors, which function as switches to activate the relays.
- A 12v power supply, powering the circuit together with a regulator and the motor.
- A 12–5v voltage regulator, regulating the voltage that reaches the microcontroller.
- Three buttons, used to send signals to change the operating mode of the microcontroller.

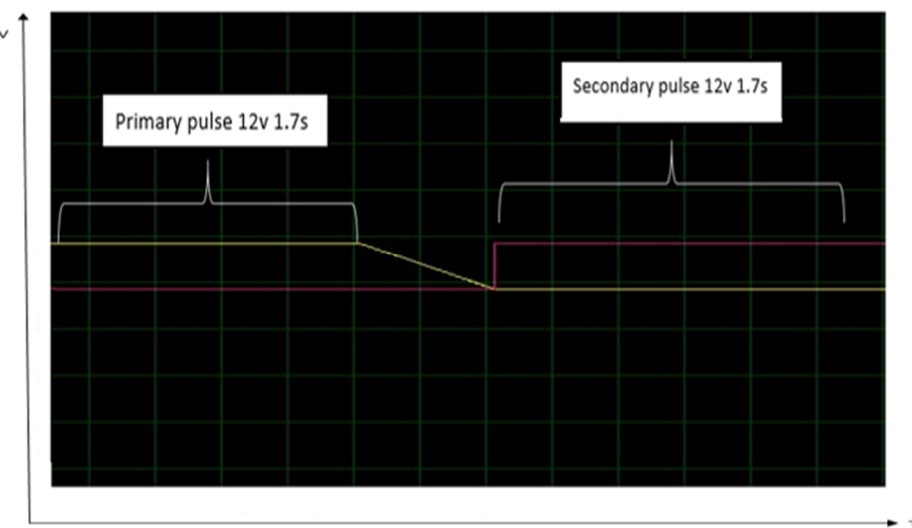

**Figure 12.** Primary and secondary pulse on stations for observing phase shift.

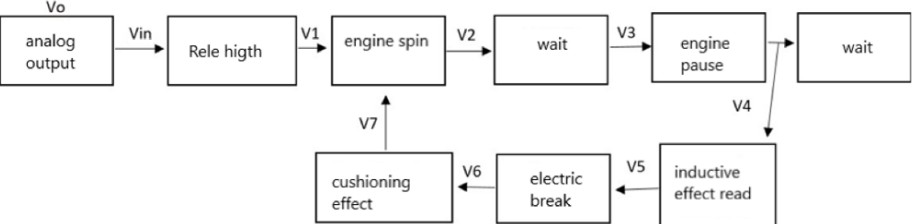

**Figure 13.** Flowchart of the operation of the 'Automatic Equipment'.

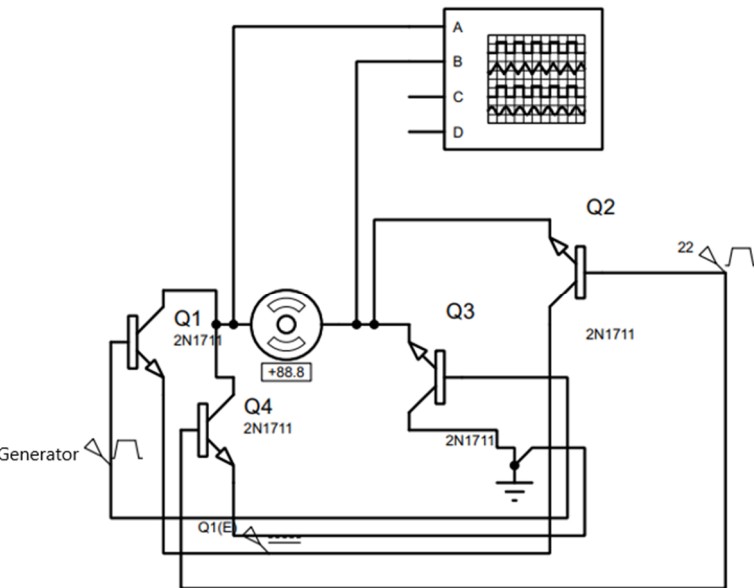

**Figure 14.** Diagram of the part of the circuit responsible for avoiding short circuits in the lines, (generator and 22 are pulses generated and sent by the microcontroller while Q1(E) is the voltage source).

The previous image, Figure 14, shows a simulation of the feasibility test of an H bridge for carrying out some of the rotation processes that we used in the automatic equipment.

### 2.3. Logical Sequence of the Software for Controlling Turns

In Figure 15, the flowchart of the software used can be seen, which controls the orientation of the current and thus can alternate the orientation of the turning; output out 1 will have negative charge initially, and output out 2 positive charge; this is achieved by activating relay 1 and 3 in parallel; to change the orientation of the current the aforementioned relays will be deactivated at the same time, and relays 2 and 4 whose wires are connected to the outputs but inversely will be activated; a feedback reading is used to avoid short circuits when changing the orientation of the current.

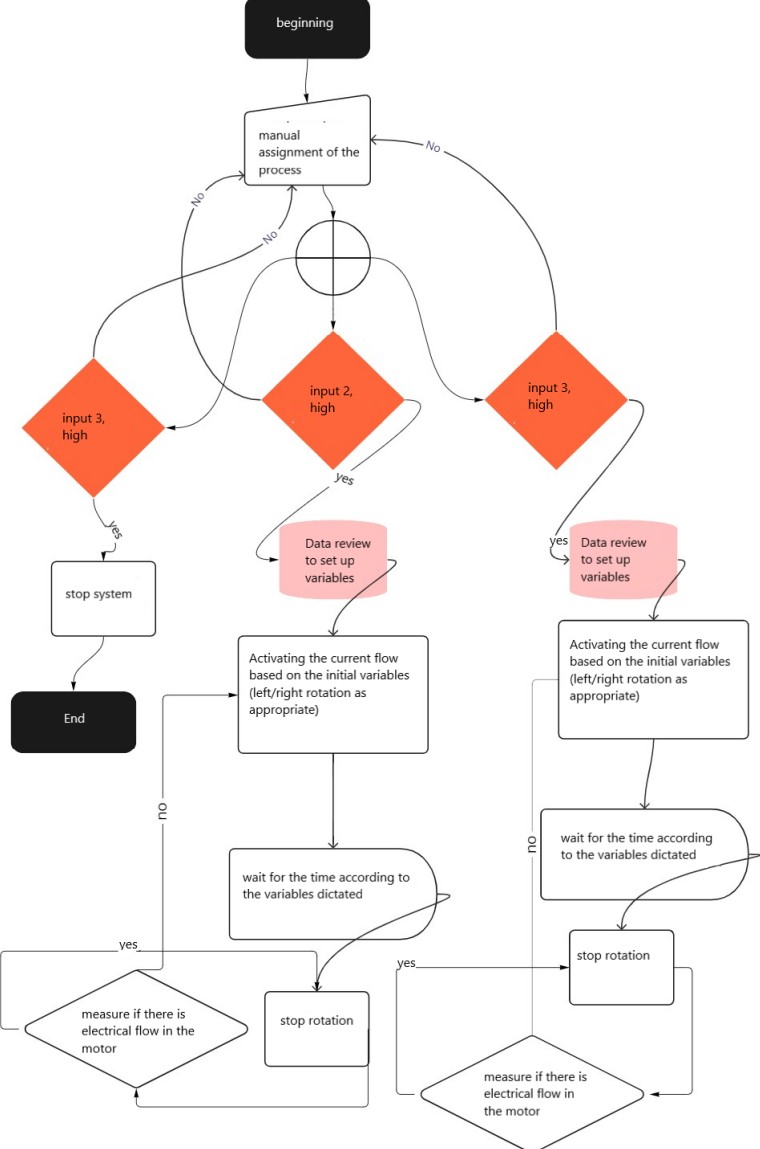

**Figure 15.** Flowchart of the software for controlling the time and rotation of the Automatic Equipment to increase sustainability in agricultural fertilization.

The software used for the control of the 'Automatic Equipment' is written in C++ language, and the Arduino IDE Version 1.0 was used.

### 2.4. Experimental Test Layout

The design of the device takes into account the resources that farmers have in the field, such as a, 200 L containers for preparing their mixtures, ammonium sulfate as a widely used fertilizer due to its low cost compared to other nitrogen fertilizers, vehicle, a tractor, etc., which with its battery can provide the power to activate a motor such as a starter

motor of the same vehicle or tractor (12 volts). In our design, from the outset, we sought to reduce the time to obtain a higher percentage of dissolved material, with less effort on the part of the farmer and with less energy consumption. To this end, we consider the dissolved fertilizer in water as a dependent variable and the time to dissolve the fertilizer in water, considering 100 kg of ammonium sulfate and 100 L of water in a 200 L tank. Our independent variables are three:

(1) An agitator: considering the traditional method used by farmers to dissolve their fertilizers in water, which corresponds to an agitator, which can be made of wood or metal, in our case we use a wooden agitator and proceed to make the intervention.

(2) Commercial mixing equipment for substances: there are many sizes, a manageable one for one person and with a power consumption available in most households (120 volts) or farms was chosen, which we know is not readily available in the field.

(3) The proposed invention, the Automatic Fertilizer Dissolving Equipment, was designed to obtain the greatest amount of fertilizer dissolved in water in a shorter time.

Initially, three drums were dissolved each with 100 L of water and 100 kg of ammonium sulfate, with three different people obtaining an average of 70 min to dissolve 90 percent of the fertilizer as shown in Table 1

**Table 1.** Comparison of results in time and efficiency in the dissolution of agricultural salts.

| Tool | Power Source | Power | Time Required to Dissolve Solid Fertilizer | Percentage of Material Dissolved |
|---|---|---|---|---|
| Automatic fertilizer dissolving equipment | D/C 12 v | 250 w | 5 min (configuration D) | 99% |
| Commercial equipment | A/C 120 v | 1200 w | 15 min | 97% |
| Manual (using a stirrer) | N/A | N/A | 70 min | 90% |

In a second stage, another three drums were dissolved each with 100 L of water and 100 kg of ammonium sulfate, with commercial mixing equipment for substances that require a 120-volt power supply and generate 1200 watts with three different people obtaining 97 percent of dissolved fertilizer with an average of 15 min as shown in Table 2.

**Table 2.** Times obtained for dissolving 100 kg of granular ammonium sulfate in 100 L of water.

| Right Turn Time | Turn Speed | Centripetal Acceleration of Turn | Left Turn Time | Time o Dissolve Fertilizer | Pause Time between Turns | Turn Configuration Identification | Reynolds Number |
|---|---|---|---|---|---|---|---|
| 4 s, 2 s | 20 m/s | 13 m/s$^2$ | 4 s, 2 s | 10–12 min | 10 s | A | 69,790 |
| 6 s, 2 s | 15 m/s | 7.5 m/s$^2$ | 6 s, 2 s | 10–12 min | 6 s | B | 52,342.5 |
| 8 s, 2 s | 10 m/s | 3.3 m/s$^2$ | 8 s, 2 s | 6–10 min | 4 s | C | 34,895 |
| 10 s, 3 s | 5 m/s | 0.8 m/s$^2$ | 10, 3 s | 5 min | 3 s | D [1] | 17,447.5 |

[1] The best-known method.

Based on these results, we set out to improve the times for dissolving the fertilizer and increasing the amount of dissolved fertilizer. To do this, we carried out several tests with the automatic fertilizer dissolving equipment as shown in the results section in Table 2, adjusting the rotation time, rotation speed, centripetal acceleration of the rotation, rotation direction, pause rotation time, and measuring the amount of Reynolds and the dependent variables until we achieved the goal of reducing the effort to dissolve a higher percentage of fertilizer in a shorter time and with less energy consumption.

The Reynolds number is used to determine the best scenario, as a higher speed spin can cause the mixture to spill due to the tangential speed.

## 3. Results

Table 1 shows the times obtained for dissolving granular Ammonium Sulfate in a 200 L container of the automatic equipment for dissolving solid fertilizers. The times in this table were obtained for dissolving 100 kg of granular ammonium sulfate in 100 L of water

in a 200 L container at environment temperature of 27°. It is important to take into account the time in which the blades are rotating and at what speed and not just the variations in the flow.

The best results were achieved by varying the spin and speed, the time between spins, and the rest time as follows: (10 s spin, 3 s rest) duration of turns to the right, (5 m/s) speed of spins, (0.8 m/s$^2$) centripetal acceleration experienced, (10 s turn, 3 s rest) duration of turns to the left, (5 min) time it takes to dissolve the salts, and (3 s) pause time between turns, and with this, we obtained a 17,447.5 Reynolds number.

Table 2, we appreciate the results of the time it takes to dissolve 100 kg of ammonium sulfate in 100 L of water with the automatic equipment for dissolving solid fertilizers, with a commercial stirrer and manually using granular ammonium sulfate as fertilizer.

As mentioned before, the invention works with laminar and turbulent flows, which help to dissolve the solute in a shorter time because a greater friction is generated with the solute than would be generated by simply subjecting the solute to a low-speed laminar flow.

$$\text{Re} = \frac{\text{pVL}}{\mu} \tag{1}$$

Symbology:
Re: Reynolds number
p: density of the liquid
V: fluid velocity
L: diameter through which the fluid moves
μ: fluid viscosity

When the Reynolds number is high, disturbances in the fluid, also called turbulent flow, occur, and conversely, when the Reynolds number is low, laminar flows occur. With the proposed invention, we have both laminar flows and turbulent flows, which is key to reducing the time it takes to achieve dissolution in a shorter time.

As can be seen in the previous Figure 16, the higher the Reynolds number, the greater the perturbations in the solvent [37], resulting in many more water particles coming into contact with the solute. However, this causes a higher speed rotation to spill the mixture due to the tangential velocity.

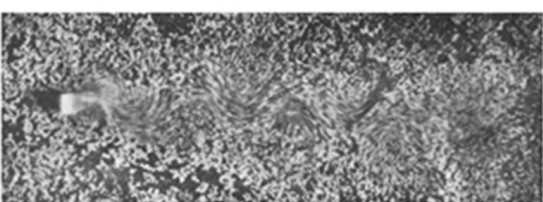
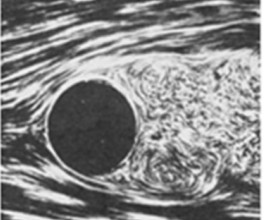

Medium number of Reynolds          High number of Reynolds

**Figure 16.** Medium and High performance of Reynolds source: [38].

The results of the fluid analysis on the blade structure indicate that the blades suffer resistance forces that range from a minimum of 20,794.68 Pa to a maximum of 187,251.09 Pa, as can be seen in Figures 17 and 18. The software used for this study is SolidWorks Flow Simulation 2020.

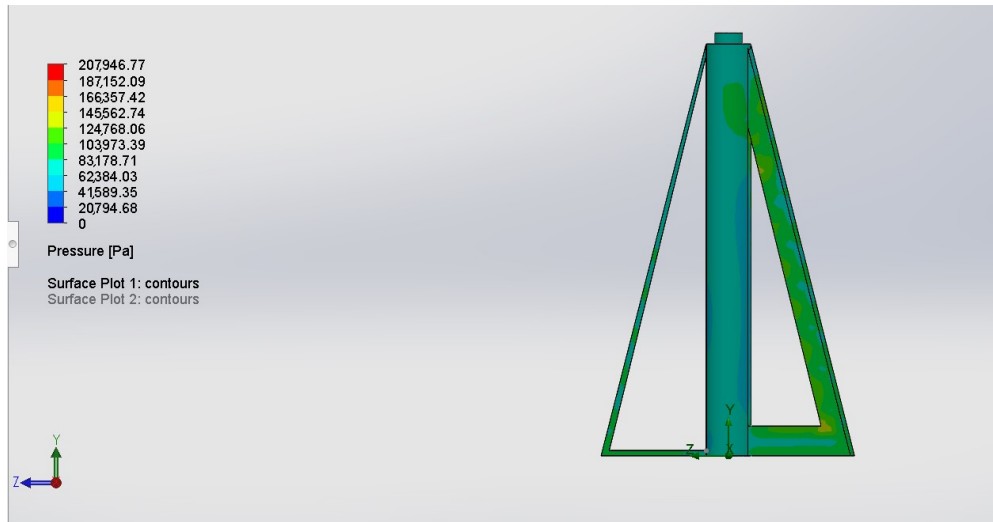

**Figure 17.** Result of pressure simulation (front view).

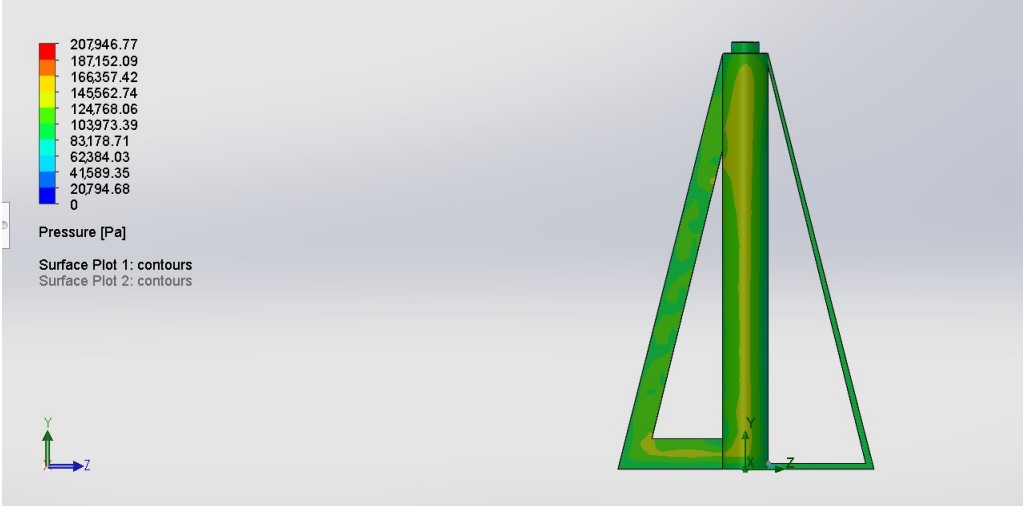

**Figure 18.** Result of pressure simulation (back view).

- Path: Liquids User Defined
- Density: 1.0267 kg/m^3
- Dynamic viscosity: 1.002 Pa*s
- Specific heat (Cp): 0.9 J/(kg*K)
- Thermal conductivity: 2.0000 W/(m*K)
- Cavitation effect: No
- Radiation properties: No

This invention can be used in the food, chemical, and any other industry where it is necessary to carry out mixing and dissolution of substances, in addition to its use to help automate the process of dissolving solid fertilizers in the field.

## 4. Discussion

Technologies that are designed address very specific needs, and their emergence is focused on supporting the reduction of effort and costs and increase sustainability in the processes where they are required [39]. Mixers are widely used in various sectors to obtain homogeneous mixes of the products to be combined; however, only standardized devices are available on the market, which do not always provide the solution sought.

In this technological development research, a very particular need in the agricultural sector was addressed: the dissolution of fertilizers, taking into account the resources available to farmers at the time they seek to carry out the dissolution for their application. In the context of peasants in the valley region of the state of Jalisco, Mexico, farmers use two-hundred-liter tanks to create their mixes and regularly prepare in the field, where there are no electric plants or contacts, and the only thing they have is a battery that can assist them in making their mixes. One of the limitations of the developed device is its size and the context for which it was developed.

On the other hand, the tests carried out showed us the values of the level of efficiency and the acceptability of the design of the proposed mixer and also demonstrated the times to obtain the mixture with respect to the traditional method used by farmers and in comparison with other devices available on the market as shown in Tables 3 and 4.

**Table 3.** Data of time saved as a result of using the invention between manual and automatic equipment for dissolving solid fertilizers.

| Duration Time for the Manual Dissolution of Salts | Duration Time for the Dissolution of Salts with the Invention | Percentage of Time Saved when Using the Invention |
|---|---|---|
| 70 min | 5 min | 92.85% |

**Table 4.** Data of time saved as a result of using the invention between a commercial mixer and the automatic equipment for dissolving solid fertilizers.

| Duration Time for the Dissolution of Fertilizer Using a Commercial Stirrer | Duration Time for the Dissolution of Fertilizer with the Invention | Percentage of Time Saved when Using the Invention |
|---|---|---|
| 15 min | 5 min | 33% |

Upon completing the tests, the invention demonstrated that using it results in a reduction in the time required for the dissolution of agricultural fertilizers, which allowed us to confirm that the invention is useful for farmers by reducing the time they have to invest in dissolving their fertilizers prior to application and effective for the dissolution of fertilizers because in a shorter time compared to commercial equipment and the traditional method; a homogeneous mixture is obtained in a shorter time and with lower energy requirements.

Finally, there was a satisfaction of the farmers with the effectiveness of the proposed invention. This coincides with various research, in which it is mentioned that the user's level of satisfaction with the proposed tool determines greater use of it [40,41]. Future research must lead us to new tools and new technology that make efficient the agriculture work and save time, money, and more profit to farmers. All the world must recognize the farmer's effort and support them as they do with the world with their work producing the food that the world needs. The farmers need technology that make their work sustainability in all ways; if they win, all of us win.

## 5. Conclusions

A device was presented. This device allows a reduction in the time to dissolve fertilizers using low energy consumption. This device has been used in fertilization processes in corn and cane in the state of Jalisco, Mexico. Farmers can use it as an alternative in liquid fertilization when weather conditions require it, in environments with little rainfall, so that crops can take advantage of nutrients immediately after fertilization and thus reduce the loss of fertilizers due to evaporation, compared to solid or granular fertilization. Therefore, by using this device in liquid fertilization, farmers optimize resources. The mechanism of the device uses a commercial transmission and motor, with specially designed paddles to dissolve fertilizers, which must be connected to a commercial 12 v vehicle battery to operate.

The paddles were specially designed to generate laminar and turbulent flows, in such a way that the energy consumption was minimized with an optimal result in the dissolution of fertilizers; with only 12 v it generates a power of 250 watts. A finger system is used at the bottom of the paddles to prevent material from being left undissolved.

The proposed mixer was tested in the fertilization of corn and cane with satisfactory results, in the laboratory, and in the field. Therefore, farmers can use this device to save money, effort, and time.

## 6. Patents

Patent pending, application number: MX/E/2022/073039.

**Author Contributions:** Conceptualization, M.M.G. and L.C.G.M.R.; methodology, S.R.C.; software, R.P.Z.; validation, M.M.G., S.R.C. and L.C.G.M.R.; formal analysis, R.P.Z.; investigation, M.M.G.; resources, L.C.G.M.R.; data curation, S.R.C.; writing—original draft preparation, M.M.G.; writing—review and editing, S.R.C.; visualization, R.P.Z.; supervision, L.C.G.M.R.; project administration, M.M.G.; funding acquisition, All authors contribute equally. All authors have read and agreed to the published version of the manuscript.

**Funding:** This research received no external funding.

**Data Availability Statement:** The data created can be appreciated in the results section.

**Conflicts of Interest:** The authors declare no conflict of interest.

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
