# Peer review of "Automatic Equipment to Increase Sustainability in Agricultural Fertilization"

_agriculture, doi:10.3390/agriculture13020490_

Round 1

Reviewer 1 Report

The paper presents the development of a new device for dissolving fertilizer in water. The prototype and the idea are interesting for the scientific topic and for the journal. However the paper needs to be extensively revised, the main issue is that the hypothesis of the work is not clearly defined and this subsequently implies that also the description of the experimental demonstration is incomplete and not reported in materials and methods. We suggest to the authors a review of the work in which they unequivocally clarify the hypothesis of the work and report the experimental plan in detail in order to demonstrate the relative hypothesis and, therefore, the values ​​reported in results.

In detail I'd suggest:

Abstract

The purpose of the study was to design a device to dissolution of fertilizers for agricultural use in an automatic and environmentally sustainable way to facilitate the work of farmers.

R. The paper doesn’t present or demonstrate the environmental sustainability of the device, also the results doesn’t measure “to facilitate the work of the farmers”. It’s important to focus the purpose of the work because it will affect all the paper structure.

Introduction 25

26-28

This article presents an automatic equipment to dissolve agricultural fertilizers to apply them as foliar or directly to the soil and improve the health of the plants with nutrients can have positive effects on the health of those who consume them.

R. The article is focused on an equipment to dissolve agricultural fertilizers. The fact that fertilizers improve plants’ health is not the topic of this work, the hypothesis is not to improve the health of the plant. I suggest to delete : “and improve the health of the plants with nutrients can have positive effects on the health of those who consume them”. As previously suggested the hypothesis of the authors has to be focused on the topic of the work.

29

The design is based on a pair of found blades propelled

R. Not clear “found blades”

31-34

The main objective of the present invention is to provide a system for dissolving agricultural nutrients and/or fertilizers for their subsequent application to crops, which allows reducing the time needed to dissolve the components of the mixtures to be used.

R. If this is the objective of the work here is clearly declared and corresponds to which is reported in the paper, but is not in accordance with the previous declarations in abstract and in row 26-28. Please conform.

70-74

… as well as the invasion of Russia into Ukraine and the increase in the price of gas, the latter being essential in the production of nitrogen in the form of ammonia and subsequently dehydrated into urea, the main producers of this fertilizer in the world are Russia, China and India have imposed restrictions on their exports to ensure internal supply [30].

R. I’d suggest to the authors to remain on the technical aspect of the equipment and on the scientific approach to its development and validation.

75-81

The world needs to produce food, countries need to ensure food production for their people [31, 32], however they do little or nothing to ensure the supplies that farmers need to maintain food production, relying on imports and dependence on third parties, which puts the viability of food production at risk, and if we add to this the inefficiency in the application of nutrients such as the traditional solid and volatilized method to crops, which generates losses of fertilizers due to evaporation, plant competition, microorganisms and water wash-off in crops [33]

R. I’d suggest to the authors to remain on the technical aspect of the equipment and on the scientific approach to its development and validation.

90-95

the invention we propose helps in the dissolution process by reducing the work required by the farmer in fertilization, and by promoting liquid fertilization, nitrogen loss is reduced because liquid application allows for immediate absorption of nutrients in the soil where they are applied and in this way pollution by wash-off in rivers, lakes and lagoons is reduced, contributing to the improvement of our environment.

R. the topic of the work has been already described clearly in 31-34. Please do not change the objective or the hypothesis of the work. I’d suggest to delete this sentence.

Materials and methods

As you adopted a 2.1 for the electronic stage, I’d suggest to introduce a 2.1 paragraph for the tool development: 2.1 The tool device (or as you prefer); 2.2 the electronic stage; 2.3…

102-104

The present invention relates to an automatic equipment for dissolving agricultural salts in liquids used in the agricultural sector;

R. Is this the objective of the work? This sentence is introduction not materials and methods.

272-273

The software used for the control of the 'Automatic Equipment' to increase sustainability in agricultural fertilization is written in C++ language and the Arduino IDE Version 1.0 was used.

R. the increased sustainability is not demonstrated, please delete “to increase sustainability in agricultural fertilization”

The experimental phase is not described. In results at table 2 appear a comparison between the new system, a commercial equipment and a manual. Please add a subchapter 2.4 Experimental test layout.

It’s not possible to evaluate the results of table 2 because the experimental plan has not been described.

Also the “manual” and the “commercial” equipment are not described. It’s not clear what has been compared and the testing method.

I would suggest also the introduction of a brief state-of-the-art chapter in which the current technologies used are described from a technical point of view without the need to indicate their strengths and weaknesses in order to understand why have been chosen the manual and the commercial as reference and which is the technical innovation of the device.

Results

Please delete row 276

277-281

Table 1 shows the times obtained for dissolving granular Ammonium Sulfate in a 200 liter container of the automatic equipment for dissolving solid fertilizers; The times in this table were obtained for dissolving 100 kilograms of granular ammonium sulfate in 100 liters of water in a 200 liter container at environment temperature of 27°. It is important to take into account the time in which the blades are rotating and at what speed and not just the variations in the flow.

R. This is material and methods.

Please check 100 kg in 100 liters!

Moreover is necessary to explain, briefly, why did you chose this fertilizer and this concentration for the experimental test

286-287

The Reynolds number is used to determine the best scenario, as a higher speed spin can cause the mixture to spill due to the tangential speed.

R. This is material and methods

288-292

The best results were achieved by varying the spin and speed, the time between spins, and the rest time as follows: (10s spin, 3s rest) duration of turns to the right, (5m/s) speed of spins, (.8m/s^2) centripetal acceleration experienced, (10s turn, 3s rest) duration of turns to the left, (5min) time it takes to dissolve the salts, (3s) pause time between turns, and with this, we obtained a 17447.5 Reynolds number.

R. As previously reported, these results are reported and seems indicating the adoption of different and significant factors affecting the results that have not been described in materials and methods. It’s necessary a chapter in materials and methods as “the experimental plan” clearly indicating the variables, the repetitions, the factors...

Discussion

Please delete row 437

377-378

Finally, there was a correlation between the levels of satisfaction of the farmers and the effectiveness of the proposed invention.

This correlation is not present and demonstrated in the text. It’s possible to report an informal level of satisfaction of the farmers but not that there’s a correlation.

386

A device was presented to increase sustainability in agricultural fertilization.

R. I suggest to delete this sentence because this was not the hypothesis of the work and the sustainability of the device has not been demonstrated.

Author Response

Prof. Dr. Les Copeland

Editor-in-Chief

Journal of Agriculture

February 11, 2023

Through this, I present the answers to the comments made by reviewer number one:

Comments and Suggestions for Authors

The paper presents the development of a new device for dissolving fertilizer in water. The prototype and the idea are interesting for the scientific topic and for the journal. However the paper needs to be extensively revised, the main issue is that the hypothesis of the work is not clearly defined and this subsequently implies that also the description of the experimental demonstration is incomplete and not reported in materials and methods. We suggest to the authors a review of the work in which they unequivocally clarify the hypothesis of the work and report the experimental plan in detail in order to demonstrate the relative hypothesis and, therefore, the values ​​reported in results.

In detail I'd suggest:

Abstract

The purpose of the study was to design a device to dissolution of fertilizers for agricultural use in an automatic and environmentally sustainable way to facilitate the work of farmers.

  1. The paper doesn’t present or demonstrate the environmental sustainability of the device, also the results doesn’t measure “to facilitate the work of the farmers”. It’s important to focus the purpose of the work because it will affect all the paper structure.

Answer

"Taking the definition from the Spanish language dictionary for sustainability which says "that which, especially in ecology and economics, can be maintained for a long time without exhausting resources or causing severe harm to the environment."

The proposed invention, as seen in Table 2 of the results section, uses 10 times less energy than a commercial device. Ecologically speaking, the invention proposed does not cause severe harm to the environment and is therefore sustainable.

Regarding the aspect that we do not measure, such as the proposed invention "to facilitate the work of farmers", we do not agree. A farmer dissolves the fertilizer, without the proposed invention, using a mixer, which can be made of wood, metal, or some other material to stir the mixture, on average takes 70 minutes to obtain complete dissolution as shown in Table 2 and with the invention, it only takes 5 minutes, without having to make any effort to stir the mixture, the invention performs the work, you just have to present it in the container, secure it, connect it to a 12-volt battery and turn it on, it only does the job."

Introduction 25

26-28

This article presents an automatic equipment to dissolve agricultural fertilizers to apply them as foliar or directly to the soil and improve the health of the plants with nutrients can have positive effects on the health of those who consume them.

  1. The article is focused on an equipment to dissolve agricultural fertilizers. The fact that fertilizers improve plants’ health is not the topic of this work, the hypothesis is not to improve the health of the plant. I suggest to delete : “and improve the health of the plants with nutrients can have positive effects on the health of those who consume them”. As previously suggested the hypothesis of the authors has to be focused on the topic of the work.

 Answer

In the template of the agriculture magazine, in the introduction section, the first paragraph says word for word "should briefly place the study in a broad context and highlight why it is important", that's why the corresponding citations [1 and 2] are incorporated and emphasis is placed on the importance of fertilization (which represents a broad context) process in which the proposed invention is located.

If we do this, we would not be following the recommendations of the template, that's why we reserve making this adjustment.

29 

The design is based on a pair of found blades propelled

  1. Not clear “found blades”

Answer

To clarify this, in Figure 1 the impellers are labeled and it is a pair of "found blades" that are driven by the motor through the transmission. 

31-34

The main objective of the present invention is to provide a system for dissolving agricultural nutrients and/or fertilizers for their subsequent application to crops, which allows reducing the time needed to dissolve the components of the mixtures to be used.

  1. If this is the objective of the work here is clearly declared and corresponds to which is reported in the paper, but is not in accordance with the previous declarations in abstract and in row 26-28. Please conform.

 Answer

It is correct, The main objective of the present invention is to provide a system for dissolving agricultural nutrients and/or fertilizers for their subsequent application to crops, which allows reducing the time needed to dissolve the components of the mixtures to be used. The statements in row 26-28 correspond to the context in which the invention is used and the corresponding citations are made.

70-74

… as well as the invasion of Russia into Ukraine and the increase in the price of gas, the latter being essential in the production of nitrogen in the form of ammonia and subsequently dehydrated into urea, the main producers of this fertilizer in the world are Russia, China and India have imposed restrictions on their exports to ensure internal supply [30].

  1. I’d suggest to the authors to remain on the technical aspect of the equipment and on the scientific approach to its development and validation.

 Answer

In the template of the agriculture journal, in the introduction section, the first paragraph says word for word "should briefly place the study in a broad context and highlight why it is important". That is why the corresponding citations [25-30] are incorporated and we emphasize the problem, the same context obliges farmers to seek technical solutions. If the context were different, the most economical, available, and accessible fertilizers would exist and there would be no need to seek alternative fertilization and fertilizer dissolution.

If we do this, we would not be following the template recommendations, so we reserve the right to make this adjustment.

75-81 (6)

The world needs to produce food, countries need to ensure food production for their people [31, 32], however they do little or nothing to ensure the supplies that farmers need to maintain food production, relying on imports and dependence on third parties, which puts the viability of food production at risk, and if we add to this the inefficiency in the application of nutrients such as the traditional solid and volatilized method to crops, which generates losses of fertilizers due to evaporation, plant competition, microorganisms and water wash-off in crops [33]

  1. I’d suggest to the authors to remain on the technical aspect of the equipment and on the scientific approach to its development and validation.

 Answer

In the template of the agriculture journal, in the introduction section, the first paragraph says word for word "should briefly place the study in a broad context and highlight why it is important". That is why the corresponding citations [31-33] are incorporated and we emphasize the problem, the same context obliges farmers to seek technical solutions. If the context were different, the most economical, available, and accessible fertilizers would exist and there would be no need to seek alternative fertilization and fertilizer dissolution.

If we do this, we would not be following the template recommendations, so we reserve the right to make this adjustment.

90-95

the invention we propose helps in the dissolution process by reducing the work required by the farmer in fertilization, and by promoting liquid fertilization, nitrogen loss is reduced because liquid application allows for immediate absorption of nutrients in the soil where they are applied and in this way pollution by wash-off in rivers, lakes and lagoons is reduced, contributing to the improvement of our environment.

  1. the topic of the work has been already described clearly in 31-34. Please do not change the objective or the hypothesis of the work. I’d suggest to delete this sentence.

Answer

The sentence was deleted.

Materials and methods

As you adopted a 2.1 for the electronic stage, I’d suggest to introduce a 2.1 paragraph for the tool development: 2.1 The tool device (or as you prefer); 2.2 the electronic stage; 2.3…

Answer

The recommendation was taken into account and we added paragraph 2.1 to the development section of the tool. 

102-104

The present invention relates to an automatic equipment for dissolving agricultural salts in liquids used in the agricultural sector;

  1. Is this the objective of the work? This sentence is introduction not materials and methods.

Answer

The sentence was erased

272-273

The software used for the control of the 'Automatic Equipment' to increase sustainability in agricultural fertilization is written in C++ language and the Arduino IDE Version 1.0 was used.

  1. the increased sustainability is not demonstrated, please delete “to increase sustainability in agricultural fertilization”

Answer

The sentence was erased

The experimental phase is not described. In results at table 2 appear a comparison between the new system, a commercial equipment and a manual. Please add a subchapter 2.4 Experimental test layout.

It’s not possible to evaluate the results of table 2 because the experimental plan has not been described.

Also the “manual” and the “commercial” equipment are not described. It’s not clear what has been compared and the testing method.

Answer

Subchapter 2.4 Experimental test layout was added.

I would suggest also the introduction of a brief state-of-the-art chapter in which the current technologies used are described from a technical point of view without the need to indicate their strengths and weaknesses in order to understand why have been chosen the manual and the commercial as a reference and which is the technical innovation of the device.

Answer

A paragraph was added in the introduction starting from line 91-104 with the mentioned suggestions.

Results

Please delete row 276

 Answer

The row was deleted

277-281

Table 1 shows the times obtained for dissolving granular Ammonium Sulfate in a 200 liter container of the automatic equipment for dissolving solid fertilizers; The times in this table were obtained for dissolving 100 kilograms of granular ammonium sulfate in 100 liters of water in a 200 liter container at environment temperature of 27°. It is important to take into account the time in which the blades are rotating and at what speed and not just the variations in the flow.

  1. This is material and methods.

Answer

The information was added to the Materials and Methods section in 2.4, Table 1 was kept in Results because it shows the results.

Please check 100 kg in 100 liters!

Answer

We check, It is correct.

Moreover is necessary to explain, briefly, why did you chose this fertilizer and this concentration for the experimental test

 Answer

It was added to subchapter 2.4.

286-287

The Reynolds number is used to determine the best scenario, as a higher speed spin can cause the mixture to spill due to the tangential speed.

  1. This is material and methods

 Answer

It was added to subchapter 2.4.

288-292

The best results were achieved by varying the spin and speed, the time between spins, and the rest time as follows: (10s spin, 3s rest) duration of turns to the right, (5m/s) speed of spins, (.8m/s^2) centripetal acceleration experienced, (10s turn, 3s rest) duration of turns to the left, (5min) time it takes to dissolve the salts, (3s) pause time between turns, and with this, we obtained a 17447.5 Reynolds number.

  1. As previously reported, these results are reported and seems indicating the adoption of different and significant factors affecting the results that have not been described in materials and methods. It’s necessary a chapter in materials and methods as “the experimental plan” clearly indicating the variables, the repetitions, the factors...

 Answer

It was added to subchapter 2.4.

Discussion

Please delete row 437

Answer

It was deleted

377-378

Finally, there was a correlation between the levels of satisfaction of the farmers and the effectiveness of the proposed invention.

This correlation is not present and demonstrated in the text. It’s possible to report an informal level of satisfaction of the farmers but not that there’s a correlation.

Answer

Thank you, it was corrected.

386

A device was presented to increase sustainability in agricultural fertilization.

  1. I suggest to delete this sentence because this was not the hypothesis of the work and the sustainability of the device has not been demonstrated.

Answer

Thank you, it was corrected.

Thank you for your consideration!

Sincerely,

Mario Martínez García, PhD

Professor, Department of Computer Science

University of Guadalajara

Reviewer 2 Report

1The parts/components in Figures 1 to 4 should be marked and explained as shown in Figure 5.

2Figure 7 is actually included in Figure 6 and should be deleted.

3Figures 8 to 12 should be appropriately cut into certain shape.

4In 3 Resultsand 4 Discussion, ”This section may be divided by subheadings. It should provide a concise and precise” ” Authors should discuss the results and how they can be interpreted from the per” should be deleted.

5In Figure 6, RL1 needs Q1 drive, while RL2... RL5 are directly driven by port C. Are you sure that port C can drive them directly?

Author Response

Prof. Dr. Les Copeland

Editor-in-Chief

Journal of Agriculture

February 11, 2023

Through this, I present the answers to the comments made by reviewer number two:

Comments and Suggestions for Authors

1、The parts/components in Figures 1 to 4 should be marked and explained as shown in Figure 5.

R. The suggestions were made and incorporated into the new version.

2、Figure 7 is actually included in Figure 6 and should be deleted.

R. The intention was to simplify the explanation of the design with the two images, however, the suggestion is considered appropriate and figure 6 was removed and adjustments were made in the numbering of all subsequent figures.

3、Figures 8 to 12 should be appropriately cut into certain shape.

R. The adjustments were made and the figures from 8 to 13 now have a uniform form.

4、In “3 Results” and “4 Discussion”, ”This section may be divided by subheadings. It should provide a concise and precise” ” Authors should discuss the results and how they can be interpreted from the per” should be deleted. 

R. The suggestions were made, and the paragraphs corresponding to the template that was not previously deleted have now been deleted.

5、In Figure 6, RL1 needs Q1 drive, while RL2... RL5 are directly driven by port C. Are you sure that port C can drive them directly?

R. yes, can drive it, because from RL2 to RL5 they have a 5-volt coil, and yet RL1, the coil is 10 volts, that's why RL1 requires Q1.

Thank you for your consideration!

Sincerely,

Mario Martínez García, PhD

Professor, Department of Computer Science

University of Guadalajara

Round 2

Reviewer 1 Report

The authors have improved the manuscript as suggested.